

**An emergent transition time-scale in the atmosphere and its implications to global-averaged precipitation control mechanisms, time-series reconstruction and stochastic downscaling**

**Miguel Nogueira\***

Instituto Dom Luiz, Faculdade de Ciências da Universidade de Lisboa

* corresponding author email: mdnogueira@fc.ul.pt

**Abstract**

Detrended Cross-Correlation Analysis (DCCA) revealed an emergent transition in non-periodic (deseasonalized) atmospheric variability at time-scales ~1-year. At multi-year time-scales (i) $\rho_{SST,Tland}$~0.6 (i.e. the correlation been global-averaged sea surface temperature, SST, and 2-meter air temperature averaged over global-land, $T_{land}$); (ii) Clausius-Clapeyron relationship becomes the dominant control of global-averaged precipitable water vapor (W), with $\rho_{W,T2m} \approx \rho_{W,SST}$~0.9; (iii) atmospheric radiative fluxes, specifically the surface downwelling longwave radiative flux (DLR), become a key constraint for global-mean precipitation (P) variability ($\rho_{P,Ratm} \approx \rho_{P,DLR}$~-0.8); (iv) cloud effects are negligible in (iii), and clear-sky DLR becomes a dominant P constraint; and (v) $\rho_{P,T2m}$ and $\rho_{P,SST}$ displayed significant multi-year correlations, although with large spread amongst different datasets (~0.4 to ~0.7). Result (v) provides a new perspective into the well-known uncertainties climate models associated with the dynamical component of precipitation. At sub-yearly time-scales all correlations underlying these five results decrease abruptly towards negligible values.

The relevance and validity of this multi-scale structure is demonstrated by three reconstructed P time-series at 2-year resolution, two relying on clear-sky DLR constraints and one based on P-SST correlation. These simple models, particularly one based on clear-sky DLR, were able to reproduce observed P anomaly time-series with similar accuracy to a (uncoupled) atmospheric model (ERA-20CM) and two climate reanalysis (ERA-20C and 20CR). The idealized models aren't applicable at sub-yearly time-scales, where the underlying correlations become negligible. However, monthly P probability density functions (PDFs) were derived by stochastic downscaling of reconstructed P,



leveraging on scale-invariant properties, outperforming the statistics simulated by ERA-
20C, 20CR and ERA-20CM.


**1.    Introduction**
The precipitation response to changes in increased concentrations of greenhouse gases is
a central topic for the climate science community. Although its regional variability is
essential to determine the societal impacts, global-averaged precipitation (P) is an
important first-order climate indicator, and a measure of the global water cycle, that must
be accurately simulated if robust climate projections are to be obtained across a wide
range of spatial and temporal scales. However, even the long-term P response is still
poorly understood, constrained and simulated (Collins et al., 2013; Allan et al., 2014;
Hegerl et al., 2015), largely due to the limited knowledge on the complex interactions
between the key components of the atmospheric branch of the water cycle and its forcing
mechanisms. This problem is tackled here by employing a multi-scale analysis framework
to study the variability of P, and its relation to two key governing mechanisms: the
Clausius-Clapeyron (C-C) relationship and the constraints imposed by the atmospheric
energy balance.
The C-C relationship is a well-known mechanism controlling the variability of the global
water cycle. Assuming constant relative humidity, it implies that fractional changes in
global-averaged precipitable water vapor ($\Delta W/W$) are linearly related to fluctuations of
global-averaged near-surface (e.g. 2-meter) air temperature ($\Delta T_{2m}$) (e.g. Held & Soden,
2006; Schneider et al., 2010):
$$\frac{\Delta W}{W} \approx \alpha_{W,T_{2m}} \Delta T_{2m},$$    (1)
where $\alpha_{W,T_{2m}} \approx 0.07$ K$^{-1}$ at temperatures typical of the lower troposphere. Numerous
studies have provided a robust confirmation for C-C at multi-decadal to centennial time-
scales, while also reporting an analogous linear response of $\Delta P$ to $\Delta T$ (see e.g. Schneider
et al., 2010; Trenberth, 2011; O'Gorman et al., 2012; and Allan et al., 2014 for reviews).
In general, these previous investigations agree on the ~7%/K sensitivity coefficient for
W. However, there is large spread on the P sensitivity coefficient estimates, typically in
the 1%/K to 3%/K range.
A widely recognized explanation for the sub-C-C sensitivity of P to temperature
fluctuations at long temporal scales comes from the atmospheric energy balance (Allen



& Ingram, 2002; Stephens & Ellis, 2008; Stephens & Hu, 2010). Specifically, averaging
over the global atmosphere, the latent heat flux associated with precipitation formation
($L_V$P, $L_V$ being the latent heat of vaporization) must be in balance with the net atmospheric
radiative flux ($R_{atm}$) and the surface sensible flux ($F_{SH}$):
$L_V P + R_{atm} + F_{SH} \approx 0,$                                                    (2)
Equation (2) represents a general state of radiative convective equilibrium (Pauluis &
Held, 2002), with energy fluxes defined positive for atmospheric gain, and negative
otherwise.
If the C-C relationship was the dominant mechanism controlling the response of
atmospheric moisture content and the global water cycle to temperature fluctuations, then
W and P could be expected to be strongly correlated to surface temperature. Previously
Gu and Adler (2011, 2012) found strong correlations between the inter-annual variability
of W and global-averaged surface temperature, in tight agreement with the C-C
relationship. However, they found weaker (but significant) correlations between the inter-
annual variability of P and global-averaged surface temperature, suggesting that C-C
might not be directly extendable to global precipitation. But these results focusing on a
single temporal scale might not represent the entire picture. In fact, it is now a well-
established fact that precipitation and other relevant atmospheric variables (including
temperature, atmospheric moisture, wind, etc.) display a complex statistical structure,
with significant variability over a wide range of temporal scales, and with the possibility
of different mechanisms governing variability at different time-scales (see e.g. Lovejoy
& Schertzer, 2013 for a comprehensive review). Furthermore, it has been shown that this
complex multiscale structure plays a role (at least) as important and the large amplitude
periodic components, namely diurnal and seasonal cycles (Lovejoy, 2015; Nogueira,
2017a). However, our understanding of the underlying governing mechanisms at different
time-scales remains largely elusive, representing a central problem for future
improvements to climate simulation and projection.
Recently, Nogueira (2018) analyzed satellite-based observational datasets, a long Global
Climate Model simulation and reanalysis products and found a tight correlation (~0.8)
between anomaly (deseasonalized) time-series of W and global-averaged surface
temperature, which emerged at time-scales larger than ~1-2 years. In contrast, at smaller
time-scales the correlation decreased rapidly towards negligible values (<0.3). In other
words, the C-C relationship is the dominant mechanism of deseasonalized W anomalies
at multi-year time-scales, but not at sub-yearly time-scales. Nogueira (2018) also found



that the magnitude of the correlations between anomaly time-series for P and global-
averaged surface temperature was negligible at sub-yearly time-scales, while at multi-
year time-scales the results showed large spread amongst different data-sets, ranging
between negligible (<0.3) and strong (~0.8) correlation values. Building on this previous
study, here the multi-scale analysis of the mechanisms governing P variability is
extended, including the energetic constraints on P represented in Equation (2).
Additionally, a simple stochastic model is proposed to reconstruct P time-series based on
the strong correlations found at multi-year time-scales, while monthly statistics are
reproduced by employing a stochastic downscaling algorithm based on scale-invariant
symmetries of P. The manuscript is organized as follows: section 2 describes the
considered datasets and the multi-scale analysis framework; the results of multi-scale
correlation analysis on P variability are presented and discussed in section 3; in section 4
a simple idealized model is proposed for reconstruction of P variability; and finally the
main conclusions are summarized and discussed in section 5.

## 2.    Data and Methodology
### 2.1.    Data sets
Observations of P were obtained from the Global Precipitation Climatology Project
(GPCP) version 2.3 monthly precipitation dataset (Adler et al., 2003), which covers the
full globe at 2.5° resolution from 1979 to present. Gridded datasets of monthly average
surface temperatures were obtained from the Goddard Institute for Space Studies
(GISSTEMP) analysis (Hansen et al., 2010), which covers the globe at 2° resolution from
1880 to present, with the values provided as anomalies relative to the 1951-1980 reference
period. GISSTEMP blends near-surface air temperature measurements from
meteorological stations (including Antarctic stations) with a reconstructed SST dataset
over oceans. Observations of atmospheric radiative fluxes were obtained from the
National Aeronautics and Space Administration (NASA) Clouds and the Earth's Radiant
Energy System, Energy Balanced and Filled (CERES-EBAF) Edition 4.0 (Loeb et al.,
2009), a monthly dataset covering the full globe at 1° resolution from March/2000 to
June/2017.
Two state-of-the-art reanalyses of the twentieth-century were considered in the present
study. One was the National Oceanic and Atmospheric Administration Cooperative
institute for Research in Environmental Sciences (NOAA-CIRES) twentieth-century
reanalysis (20CR) version 2c (Compo et al., 2011), which covers the full globe at 2°



resolution, spanning from 1851 to 2014. Only surface pressure observations and reports
are assimilated in this reanalysis. SST boundary conditions are obtained from 18 members
of pentad Simple Ocean Data Assimilation with Sparse Input (SODAsi) version 2, with
the high latitudes corrected to the Centennial in Situ Observation-Based Estimates of the
Variability of SST and Marine Meteorological Variables, version 2 (COBE-SST2). Here,
global-mean time-series of P, W, SST, $T_{2m}$, DLR and OLR are obtained from 20CR at
daily resolution for the 1900-2010 period. $R_{atm}$ cannot be obtained the incoming solar
radiation at TOA is not available for the 20CR dataset, due to an error with output
processing.
The other reanalysis considered in the present study was the European Centre for
Medium-Range Weather Forecasts (ECMWF) twentieth-century reanalysis (ERA-20C,
Poli et al., 2015), which covers the full globe at 1° resolution spanning from 1900-2010.
It assimilates marine surface winds from the International Comprehensive Ocean-
Atmosphere Data Set version 2.5.1 (ICOADSv2.5.1) and surface and mean-sea-level
pressure from the International Surface Pressure Databank version 3.2.6 (ISPDv3.2.6)
and from ICOADSv2.5.1. SST boundary conditions are obtained from the Hadley Centre
Sea Ice and Sea Surface Temperature data set version 2.1 (HadISST2.1). Global-mean
time-series of P, W, SST, $T_{2m}$, $R_{atm}$, DLR and OLR are obtained from ERA-20C at daily
resolution for the 1900-2010 period.
Finally, the uncoupled ECMWF twentieth-century ensemble of ten atmospheric model
integrations (ERA-20CM, Hersbach et al., 2015) was considered, which uses the same
model, grid, initial conditions, radiative and aerosol forcings as ERA-20C. However, no
observations are assimilated, the simulation is integrated continuously over the full 1900-
2010 period, and SST is prescribed by an ensemble of realizations from HadISST2.1,
including one control simulation and nine simulations with perturbed SST and sea-ice
concentration. A 10-member ensemble of global-mean time-series of P, W, SST, $T_{2m}$,
$R_{atm}$, DLR and OLR were obtained from ERA-20CM at monthly resolution for the 1900-
2010 period. Considering ERA-20CM allowed to test the sensitivity of the multi-scale
correlation structure derived from ERA-20C to data assimilation, but different
atmospheric evolutions associated with perturbations to the forcing fields (particularly to
SST).
Notice that the clear-sky radiative fluxes considered here obtained from ECMWF datasets
are computed for the same atmospheric conditions of temperature, humidity, ozone, trace
gases and aerosol, but assuming that the clouds are not there. Clear-sky estimates from





ERA-20C and ERA-20CM cover the full globe area and not just the cloud free regions at
each time instant. However, they are available for net radiative fluxes, but not for
downwelling or upwelling radiation fluxes.

## 2.2. Detrended Cross-Correlation Analysis (DCCA)

DCCA allows to accurately quantify power-law correlations between two different time-
series over wide ranges of time-scales (Podobnik & Stanley, 2008). Consider two time-
series, $y$ and $y'$, with N data points each. Due to the strong yearly cycle present in the
considered time-series, the periodic seasonal trend is first eliminated by subtracting the
long-term average (over all the years in the record) of each calendar day (or month,
depending on temporal resolution):
$\Delta y(i) = y(i) - \langle y \rangle_d,$        (3)
Then two integrated signals, $R$ and $R'$, are constructed from the deseasonalized anomaly
time-series, $\Delta y$ and $\Delta y'$:
$R_k = \sum_{i=1}^{k}[\Delta y(i) - \langle y_{ds} \rangle],$        (4)
Where k=1,…,N and $\langle\ \rangle$ indicates temporal averaging. The integrated signals are
divided into $N - n$ overlapping segments, each containing $n + 1$ values. For each
segment from each integrated signal, the "local trend" is estimated using a first-order
polynomial. The detrended integrated signal is then defined as the difference between the
original integrated signal and the local trend $(R_v - \widetilde{R_v})$, where $\widetilde{R_v}$ is the fitting first-order
polynomial to the $v$th segment $R_v$. Next, the covariance of the residuals in each segment
is calculated as:
$f_{R,R'}{}^2(n,i) = \frac{1}{n+1}\sum_{k=i}^{i+n}[(R_v - \widetilde{R_v})(R_v' - \widetilde{R_v}')],$        (5)
The detrended covariance is estimated by summing over all overlapping N-n segments:
$F_{R,R'}^2(n) = \frac{1}{N-n}\sum_{i=1}^{N-n} f_{R,R'}^2(n,i),$        (6)
Finally, the DCCA cross-correlation coefficient at time-scale $n$, $\rho_{y,y'}(n)$, is defined as
the ratio between the detrended covariance function and the product of the square-rooted
detrended variance function for each time-series:
$\rho_{y,y'}(n) = \frac{F_{R,R'}^2(n)}{\sqrt{F_{R,R}^2(n)} \times \sqrt{F_{R',R'}^2(n)}},$        (7)
The values of $\rho_{y,y'}(n)$ range between -1 and 1 (for perfect negatively and positively
correlated signals, respectively). It has been previously shown that critical points for the
95% significance level of $|\rho_{DCCA}|$ can vary between values below 0.1 and up to about 0.4,




depending on the time series length, the considered time-scale, and the power law
exponents of both time-series (Podobnik et al., 2011). Here it is assumed that $|\rho_{DCCA}|$
values below 0.3 are nonsignificant, and that $|\rho_{DCCA}|$ values in the 0.3 to 0.4 range should
be interpreted with care.

**3.    DCCA analysis of the mechanisms governing P variability across time-scales**
3.1.  **Multi-scale structure of the atmospheric water cycle response to surface**

**temperature fluctuations**

DCCA reveals strong correlations (~0.9) between deseasonalized anomaly time-series for
W and $T_{2m}$ or SST at multi-year time-scales (Fig. 1a). However, as the time-scale
decreases there is a transition in the correlation structure, and negligible correlations
(<0.3) emerge at sub-yearly time-scales. This result suggested that the C-C relationship
in Equation (1) holds to a very good approximation at multi-year time-scales, but not at
sub-yearly time-scales. Lovejoy et al. (2018) employed multi-scale analysis framework
based on Haar wavelets to GISSTEMP and found a similar transition in the multi-scale
correlation structure of SST against global-averaged surface temperature, between low-
correlations at time-scales below a few months and strong correlations (~0.8) at multi-
year time-scales. These strong correlations weren't surprising, since SST was a major
component in their definition of global-averaged surface temperature (also considering
SST over the ocean pixels and 2-meter air temperature over land pixels). But their results
also showed a transition in the correlation coefficients between SST and near-surface air
temperature over global-land ($T_{land}$), with maximum correlation values ~0.6 at multi-year
time-scales. The transition in $\rho_{SST,Tland}$ was confirmed here by employing DCCA to
ERA-20C, ERA-20CM, 20CR and GISSTEMP (Fig. 1b). Thus, the present results
support Lovejoy et al. (2018) argument that these abrupt correlation changes correspond
to a fundamental behavioral transition, where the atmosphere and the oceans start to act
as a single coupled system. Furthermore, the results presented here suggest that W
anomalies at multi-year resolution can be derived, to a very good approximation, from
SST alone.
Nogueira (2018) also reported a transition in the multi-scale correlation structure between
deseasonalized anomaly time-series of P and global-averaged surface temperature
(considering SST over the oceans and $T_{2m}$ over land), with negligible values at sub-yearly
time-scales, but with large spread in the magnitude of the multi-year correlations, ranging



between values ~0.3 to ~0.8. Here, a similar result was found for $\rho_{P,T_{2m}}$ and $\rho_{P,SST}$ (Fig.
1c), with large spread in correlation magnitude at multi-year time-scales (~0.7 in ERA-
20C and ERA-20CM, ~0.6 in 20CR, and <0.4 in observations). This large spread and the
relatively low correlations obtained from observational datasets confirmed the
uncertainty on the extension of C-C relationship as the dominant control of P variability.
Notice that the large spread in $\rho_{P,T_{2m}}$ and $\rho_{P,SST}$ represents a different perspective, under
a multi-scale analysis framework, on a previously established fact: there are large
uncertainties in climate simulations associated with the role of the non-thermodynamical
(circulation) component of precipitation response to climate change (see e.g. Shepherd,

2014).

**3.2. Multi-scales structure of the energetic constraints to P variability**

A study of the circulation component of the P response to temperature fluctuations
requires a detailed representation of several spatially heterogeneous variables and their
nonlinear interactions. An alternative path towards understanding P variability was taken
in the present investigation, focusing on the constraints imposed by the atmospheric
energy balance represented in Equation (2). Fig. 2a (solid lines) shows that the estimated
DCCA correlation coefficients between the deseasonalized anomaly time-series for P and
$R_{atm}$ were strongly (negatively) correlated at multi-year time-scales ($\rho_{P,R_{atm}} \sim -0.8$ in
ERA-20C, ERA-20CM and observations), in agreement with the balance in Equation (2).
The same wasn't true at sub-yearly time-scales, where the correlation magnitude
decreased rapidly, changed sign around monthly time-scales, and reached values ~0.4 at
time-scales below about 10 days.
Considering the effect of $F_{SH}$ in Equation (2) (i.e. $\rho_{P,R_{atm}+F_{SH}}$) slightly increased the
(positive) correlations at sub-monthly time-scales (Fig. 2a, dashed lines), although the
absolute changes are essentially below 0.1 and $\rho_{P,R_{atm}+F_{SH}}$ at sub-monthly time-scales
(which is only available for the ERA-20C dataset). More importantly, the change between
$\rho_{P,R_{atm}}$ and $\rho_{P,R_{atm}+F_{SH}}$ at multi-year time-scales was negligible. Indeed,
$\rho_{P,R_{F_{SH}}}$ displayed values up to about 0.5 at sub-monthly time-scales, but essentially <0.4
at multi-year time-scales (Fig. 2a, dot-dashed lines). Given the results in Fig. 1a, the
following linear relation was hypothesized: $L_V \Delta P \approx c_1 \times (-\Delta R_{atm}) + c_2$, where $c_1$ and
$c_2$ are arbitrary constants, and $\Delta$ represents fluctuations taken as deseasonalized
anomalies at multi-year resolutions. At sub-yearly time-scales this simplification is not





adequate, since $\rho_{P,R_{atm}}$ becomes negligible and, thus, the energy balance represented in
Equation (2) doesn't represent the dominant constraint on P variability, most likely due
to non-negligible changes in atmospheric heat storage.
The analysis was extended by decomposing $R_{atm}$ into its net atmospheric longwave and
shortwave radiative flux components, i.e. $R_{atm} = R_{LW,net} + R_{SW,net}$. On the one hand,
$\rho_{P,R_{atm}} \approx \rho_{P,R_{LW,net}}$ over the full range of time-scales considered (Fig. 2b). On the other
hand, $\rho_{P,R_{SW,net}}$ also displays significant values (~0.6) at multi-year time-scales, but the
latter magnitude was nearly 0.2 lower than $\rho_{P,R_{atm}}$ and $\rho_{P,R_{LW,net}}$ (Fig. 2b). Consequently,
the above linear relationship for multi-scale P anomalies was further refined as $L_V \Delta P \approx$
$c_1 \times (-\Delta R_{atm}) + c_2 \approx c_3 \times (-\Delta R_{LW,net}) + c_4$, where $c_3$ and $c_4$ are arbitrary constants.
Subsequently, $R_{LW,net}$ was further decomposed into the top-of-atmosphere (TOA) and
surface net longwave fluxes, i.e. $R_{LW,net} = R_{LW,TOA} + R_{LW,SFC}$. At multi-year time-
scales, $\rho_{P,R_{atm}} \approx \rho_{P,R_{LW,SFC}}$ (Fig. 2c). $\rho_{P,R_{LW,TOA}}$ also displayed significant values at
multi-year time-scales, up to ~-0.6 in ERA-20C and ERA-20CM datasets. Notice that
20CR displayed values $\left| \rho_{P,R_{LW,TOA}} \right| < 0.4$ at multi-year time-scales. But ECMWF
datasets were in better agreement with observations, suggesting that significant (negative)
correlations existed between P and $R_{LW,TOA}$ anomalies at multi-year time-scales.
Nonetheless, even for ECMWF and observational products, the magnitude of $\rho_{P,R_{LW,TOA}}$
at multi-year time-scales was nearly 0.2 lower than for $\rho_{P,R_{LW,SFC}}$. Consequently, a further
approximation was considered on the linear model for P fluctuations at multi-year time-
scales:     $L_V \Delta P \approx c_1 \times (-\Delta R_{atm}) + c_2 \approx c_3 \times (-\Delta R_{LW,net}) + c_4 \approx c_5 \times$
$(-\Delta R_{LW,SFC}) + c_6$.
Finally, $R_{LW,SFC}$ can be further decomposed into its upwelling ($R_{LW,SFC,UP}$) and
downwelling ($R_{LW,SFC,DOWN}$, henceforth denoted downwelling longwave radiation, DLR)
components. Fig. 2d shows that, at multi-year time-scales, the differences between
$\rho_{P,R_{DLR}}$ and $\rho_{P,R_{atm}}$ were within 0.1 in observations, ERA-20C and ERA-20CM ($R_{atm}$ is
unavailable for 20CR). Thus, at multi-year time-scales, the fluctuations in downwelling
surface longwave radiative fluxes are, to a good approximation, linearly related to P
fluctuations: $L_V \Delta P \approx c_7 \times (-\Delta DLR) + c_8$. Notice that the differences between
$\rho_{P,R_{LW,SFC,UP}}$ and $\rho_{P,R_{atm}}$ are identically low in observations, but these differences are
somewhat higher (~0.2) in ERA-20CM and ERA-20C. Thus, a similar linear relationship





between $\Delta P$ and $\Delta R_{LW,SFC,UP}$ might also hold to a good approximation, although the
correlations are less robust than for $\Delta P$ against $\Delta DLR$.
The correlation between global-mean clear-sky net radiative atmospheric heating and P,
i.e. $\rho_{P,R_{atm,cs}}$, was nearly identical to $\rho_{P,R_{atm}}$ at multi-year time-scales (Fig. 3a). This
suggested that the cloud effects on the multi-year linear dependence between P variability
and net atmospheric radiative fluxes may be neglected. But the same isn't true at time-
scales below a few months, where significant differences emerge between $\rho_{P,R_{atm,cs}}$ and
$\rho_{P,R_{atm}}$. This clear-sky approximation holds at multi-year time-scales for correlations of
P against global-averaged net atmospheric longwave radiative fluxes and, also, and
against the global-averaged net surface longwave fluxes (Fig. 3b). Based on these results,
it was further hypothesized that cloud effects are also negligible for the correlation
between P and DLR at multi-year temporal scales. This hypothesis could not be tested
directly because clear-sky DLR time-series were not available for the ECMWF datasets.
Nonetheless, the results in Section 4 based on an empirical algorithm for DLR estimation
under a clear-sky approximation provided support for this hypothesis.
In summary, DCCA suggested that P variability at multi-year time-scales is linearly
related to the net atmospheric radiative fluxes. Furthermore, this linear relationship is
dominated by its longwave component and, more specifically, by the surface longwave
radiative fluxes, particularly DLR. DCCA also suggests that clouds play a negligible
effect in these linear correlations at multi-years scales. The hypothesized tight correlation
between P and clear-sky DLR fluxes at multi-year time-scales was particularly
interesting, since clear-sky DLR may be estimated directly from atmospheric water vapor
content and surface temperature (e.g. Stephens et al., 2012b). This fact will be further
explored below, in Section 4.
Finally, notice that the results in Fig. 2c showed that P variability was best correlated to
$R_{LW,TOA}$ variability at sub-monthly time-scales, reaching positive values ~0.5-0.6. This
corresponds to a well-known relation between convective rainfall and the outgoing
longwave radiation at TOA, often denoted OLR (e.g. Xie & Arkin, 1998). However, this
result provided no further simplification in the sense that, unlike for clear-sky DLR at
multi-year resolution, it is equally difficult to model and predict P and OLR (including
cloud effects) at sub-monthly time-scales.
At this point, it is important to notice that the existence of strong correlations does not
necessarily imply causality between two variables. However, the atmospheric energy



balance in Equation (2) provides a physical basis for the obtained strong (negative)
correlations values between P and atmospheric radiative fluxes. In fact, the importance
of energetic constraints to global precipitation, the dominant role of surface longwave
fluxes, namely DLR, and the negligible cloud effects in these relations has been pointed
out by previous investigations (e.g., Stephens and Hu, 2010; Stephens et al., 2012a,b).
The DCCA presented here provided further robustness to these results. More importantly,
a clear transition emerged between robust correlations at multi-year time-scales and
negligible correlations at sub-yearly time-scales, which was found for P against $R_{atm}$ (or
DLR), for W against $T_{2m}$ (and SST), for SST against $T_{land}$ and, less robustly, for P against
$T_{2m}$ (or SST). Given the interdependence between these variables, these transitions are
likely to be interrelated, representing a more fundamental transition in the atmosphere.
Notice that these results also contribute to sharpen the picture from previous studies
reporting a 'fast' P response at sub-monthly time-scales, where P is suggested respond
directly to the radiative effects of increasing $CO_2$; and a 'slow' response where P increases
due to increasing surface temperature (Allen & Ingram, 2002; Bala et al., 2010; Andrews
et al., 2010; O'Gorman et al., 2012; Allan et al., 2014).

**4.    Stochastic model for global-mean precipitation**
**4.1.    Reconstruction of P time-series at multi-year resolution**
Here a very simple model for P response to climate change is proposed aiming to
demonstrate the robustness of the tight correlation between P and clear-sky DLR ($DLR_{CS}$)
at multi-year time-scales presented in Section 3. The rationale is that the correlation
between P and $DLR_{CS}$ at multi-year time-scales is significantly more robust than between
P and $T_{2m}$ (or SST). Additionally, $DLR_{CS}$ can be derived, to a good approximation, from
the global averaged near-surface temperature alone (e.g. Stephens et al., 2012b).
Furthermore, given the tight coupling between $T_{land}$ and SST at multi-year time-scales
(Fig. 1b), it is hypothesized that $DLR_{CS}$ variability could be obtained, to a good
approximation directly from the SST forcing. This hypothesis is also supported by the
nearly identical correlations between W and $T_{2m}$ or SST (Fig. 1a).
Here two different algorithms to estimate $DLR_{CS}$ are tested: the Dilley-O'Brien model
(Dilley & O'Brien, 1998), and the Prata model (Prata, 1996). In the Dilley-O'Brien
model:
$DLR_{2y,DO} = a_1 + a_2 \left(\frac{SST_{2y}}{SST_c}\right)^6 + a_3 \left(\frac{\Delta W_{2y}+W_c}{W_c}\right)^{1/2},$         (8)





Where $a_1 = 59.38\ \mathrm{Wm^{-2}}$, $a_2 = 113.7\ \mathrm{Wm^{-2}}$ and $a_3 = 96.96\ \mathrm{Wm^{-2}}$ are the model parameters,
and $W_c = 22.5\ \mathrm{kg\ m^{-2}}$ is the climatological value for W. The subscript '2y' (e.g. $DLR_{2y}$)
indicates a time-series at 2-year resolution. The fluctuations $\Delta$ represent anomaly time-
series relative to a climatological time-series, for example $\Delta DLR_{2y,DO} = DLR_{2y,DO} -$
$DLR_{c,DO}$. Notice that for multi-year resolution time-series, this yields the same result as
first deseasonalizing the time-series (using Equation (3)) and then coarse-graining it to 2-
year resolution. $DLR_{c,DO} = a_1 + a_2 + a_3$ is obtained by replacing the climatological
values of W and SST in Equation (8).
The Prata model for $\Delta DLR_{2y,Pr}$ is based on the Stefan-Boltzmann equation:
$DLR_{2y,Pr} = \varepsilon_{clr}\sigma_{SB}SST_{2y}{}^4$ (9)
Where $\sigma_{SB} = 5.67 \times 10^{-8}\ \mathrm{Wm^{-2}K^{-4}}$ is the Stefan-Boltzmann constant and:
$\varepsilon_{clr} = 1 - \left(1 + W_{2y}\right)\exp(-\left(1.2 + 3W_{2y}\right)^{1/2})$ (10)
The anomaly-time series is computed from $\Delta DLR_{2y,Pr} = DLR_{2y,Pr} - DLR_{c,Pr}$, where
$DLR_{c,Pr}$ is obtained by replacing the climatological values of W and SST in Equations
(9) and (10).
The high values of $\rho_{W,SST}(\approx \rho_{W,T_{2m}})$ at multi-year time-scales (Section 3.1) allowed to
remove the W dependence in Equations (8) and (11), by replacing    $W_{2y} \approx$
$\alpha_{W,SST}\Delta SST_{2y}W_c + W_c$. The forcing $\Delta SST_{2y}$ time-series were obtained by coarse-
graining the deseasonalized (using Equation (3)) global-averaged SST obtained from
GISSTEMP dataset. The sensitivity coefficient, $\alpha_{W,SST} \approx 0.08\ K^{-1}$ was estimated by
least-square regression of $\Delta W_{2y}/W_c$ against $\Delta SST_{2y}$, pooling together all datasets (ERA-
20C, ERA-20CM and 20CR). The $\alpha_{W,SST}$ estimates are summarized in Table 1, including
for each individual dataset, ranging between 0.07 and 0.10 K$^{-1}$. Notice that the obtained
values are close to the typical 0.07 K$^{-1}$ value.
The results from Section 3.2 suggested a linear relation between P and DLR$_{CS}$ variability
at multi-year time-scales, which can be written as $P_{2y} \approx \alpha_{P,DLR}(-\Delta DLR_{CS,2y})P_c + P_c$. In
this way, two reconstructed anomaly time-series for P were obtained, $P_{2y,DO}$ and $P_{2y,Pr}$,
respectively by replacing $\Delta DLR_{CS,2y}$ with $\Delta DLR_{2y,DO}$ and $\Delta DLR_{2y,Pr}$. The coefficient
$P_c \approx 2.7$ mm/day was estimated from GPCP dataset. The sensitivity coefficient $\alpha_{P,DLR} \approx$
0.004 (W/m$^2$)$^{-1}$ was estimated by least-square regression of $\Delta P_{2y}/P_c$ against $\Delta DLR_{2y}$,
pooling together all available datasets (ERA-20C, ERA-20CM, 20CR and GPCP against
CERES-EBAF). Notice that, in estimating $\alpha_{P,DLR}$, clear-sky DLR time-series were used





394 where available (i.e. for ERA-20C and ERA-20CM) datasets, but they were replaced by

395 (full-sky) DLR otherwise (i.e. for 20CR and CERES-EBAF). The $\alpha_{P,DLR}$ estimates are

396 summarized in Table 2, including values obtained from each dataset (no estimate was

397 made for GPCP against CERES-EBAF due to the limited duration of the latter), ranging

398 between 0.003 $(W/m^2)^{-1}$ and 0.005 $(W/m^2)^{-1}$.

399 Another simple linear model for reconstruction of multi-year P anomaly time-series was

400 tested, based on the direct response (correlations) of P to SST fluctuations, i.e. $P_{2y,SST} \approx$

401 $\alpha_{P,SST} \Delta SST_{2y} P_c + P_c$. Again, the $\Delta SST_{2y}$ was obtained from GISSTEMP dataset. The

402 sensitivity coefficient, $\alpha_{P,SST} \approx 0.02\ K^{-1}$ was estimated by least-square regression of

403 $\Delta P_{2y}/P_c$ against $\Delta SST_{2y}$, pooling together all datasets (ERA-20C, ERA-20CM, 20CR and

404 GPCP against GISSTEMP). The $\alpha_{P,SST}$ estimates are summarized in Table 3, including

405 for each individual dataset, ranging between 0.02 and 0.04 $K^{-1}$. Notice that the obtained

406 values are close to the 0.01 to 0.03 $K^{-1}$ range reported in the relevant literature (e.g.

407 Schneider et al., 2010; Trenberth, 2011; O'Gorman et al., 2012; and Allan et al., 2014).

408 When compared against $\Delta P_{2y}$ directly derived from GPCP for the 1979 to 2010 period,

409 the errors in the proposed linear $\Delta P_{2y}$ reconstructions were generally close to those for

410 atmospheric model-based products (Fig. 4). $\Delta P_{2y,Pr}$ displays the highest mean bias,

411 somewhat higher than for atmospheric model-based datasets, but also higher than the

412 mean bias for $\Delta P_{2y,DO}$ and $\Delta P_{2y,SST}$ (Fig. 4a). Notice that all atmospheric model-based

413 products considered here also display a positive bias. While this may be due a negative

414 bias of GPCP (e.g. Gehne et al., 2015), this observational dataset represents the longest

415 reliable dataset for global precipitation studies and thus was considered here as 'the truth'.

416 More importantly, the mean bias is easy to correct, simply by subtracting its value from

417 the time-series. This correction was implemented here for all atmospheric model-based

418 and linear-model based $\Delta P_{2y}$ time-series. Figure 4c shows that the normalized standard

419 deviation ($\sigma_n = \sigma_{2y,model}/\sigma_{2y,obs}$) estimated from $\Delta P_{2y,DO}$ (~0.4) and, particularly, from

420 $\Delta P_{2y,SST}$ (~0.3) were lower than the values estimated from atmospheric model-based

421 products (~0.5-0.9). In contrast, $\sigma_n$ estimated from $\Delta P_{2y,Pr}$ was nearly 0.8, which was

422 higher than 20CR and most members in the ERA-20CM ensemble, only outperformed by

423 ERA-20C dataset. The root-mean squared error after bias-correction (RMSE$_{bc}$) estimated

424 from $\Delta P_{2y,Pr}$ and $\Delta P_{2y,DO}$ were well within the range of the values obtained from

425 atmospheric model-based products (Fig. 4b), with the Prata model slightly





overperforming the Dilley-O'Brien model. $RMSE_{bc}$ estimated from $\Delta P_{2y,SST}$ was on the
high-end of the atmospheric model-based range of values, and somewhat worse than for
the DLR-based linear models. Finally, the Pearson correlation coefficient between models
and observations (Fig. 4d) was similar amongst all linear-based models and well within
the range of values estimated from the atmospheric model-based products. The latter
result was expected since all linear models were forced by the same SST time-series.
Overall, these results suggested that $\Delta P_{2y,Pr}$ (after bias correction) reproduced the
observations with similar accuracy to atmospheric model-based products, including
similar $RMSE_{bc}$, variability amplitude and phase of the signal. $\Delta P_{2y,DO}$ displayed similar
performance for $RMSE_{bc}$ and for the phase, but not for the variability amplitude. Finally,
$\Delta P_{2y,SST}$ had the worst performance concerning $RMSE_{bc}$, but also in capturing the
variability amplitude, while it displayed similar ability to the other linear models in
reproducing the phase. The overall weakest performance of $\Delta P_{2y,SST}$ was coherent with
the less robust correlations underlying this model. Additionally, the results indicate that
the non-linear transformations on SST employed in the Prata and the Dilley-O'Brien
algorithms improved the linear models.
**4.2. Stochastic reproducing of P monthly PDFs**
At sub-yearly time-scales, the magnitude of $\rho_{P,W}$, $\rho_{P,SST}$ and $\rho_{P,DLR}$ decreased abruptly
to negligible values (Section 3). Thus, at these time-scales, the C-C relationship is no
longer the dominant control of W (nor P) variability, and the longwave radiative fluxes
are no longer the main constraints for P. Additionally, the cloud-effects on P variability
become non-negligible (Fig. 3). Consequently, the linear relationships underlying the
above P reconstruction at 2-year resolution are no longer appropriate at sub-yearly time-
scales. Building on the strong scale-invariant symmetries present in the variability of
global and regional rainfall across wide ranges of time-scales (e.g. Lovejoy and Schertzer,
2013; Nogueira et al., 2013; Nogueira and Barros, 2014, 2015; Nogueira, 2017, 2018), an
algorithm was proposed here to derive the sub-yearly statistics from the multi-year
information alone. The physical basis for this algorithm is that while the atmosphere is
governed by continuum mechanics and thermodynamics, it simultaneously obeys
statistical turbulence cascade laws (e.g., Lovejoy & Schertzer, 2013; Lovejoy et al.,

2018).

Conveniently, precipitation (and many other atmospheric variables) is characterized by
low spectral slopes $\beta < 1$, with quasi-Gaussian and quasi-non-intermittent statistics, at



time-scales between ~10 days and a few decades (Lovejoy & Schertzer, 2013; de Lima
& Lovejoy, 2015; Lovejoy et al., 2015, 2018; Nogueira, 2017b, 2018). Grounded by these
scale-invariant properties, fractional Gaussian noise was used here to generate multiple
realizations of downscaled $\Delta P$ at monthly resolution from each member of each $\Delta P_{2y}$
time-series:
$$\Delta P_{1m}(t) = fGn_{1m}(t)\frac{\Delta P_{2y}(t)}{fGn_{2y}(t)} \quad\quad\quad\quad\quad (11)$$
where $fGn_{1m}$ is a fractional Gaussian noise, which was computed by first generating a
random Gaussian noise $(g)$, then taking its Fourier transform $(\tilde{g})$, multiplying by $k^{-\beta/2}$,
and finally taking the inverse transform to obtain $fGn_{1m}$. The mean of $fGn_{1m}$ was forced
to be equal to the number of data-points of $\Delta P_{2y}$. Then $fGn_{2y}$ was obtained by coarse-
graining $fGn_{1m}$ using 24-point (i.e. 2 years) length boxes. In this way, $\Delta P_{1m,DO}$, $\Delta P_{1m,Pr}$,
$\Delta P_{1m,SST}$ ensembles are generated respectively from the bias-corrected $\Delta P_{2y,DO}$, $\Delta P_{2y,Pr}$
and $\Delta P_{2y,SST}$ time-series. One hundred plausible realizations are generated for each
ensemble, corresponding to one hundred different realizations of $fGn_{1m}$. Based on recent
investigations on P scale-invariance properties, a spectral exponent $\beta \approx 0.3$ is assumed
(de Lima & Lovejoy, 2015; Nogueira, 2018). In Equation (11), the 2-year resolution time-
series were assumed to have a constant value for every month inside each 2-years period.
Notice that a resolution limit should exist to the proposed stochastic downscaling
algorithm, namely at time-scales below ~10 days where a fundamental transition occurs
in the scaling behavior of most atmospheric fields (including P, see e.g. Lovejoy &
Schertzer, 2013; Lovejoy, 2015; de Lima & Lovejoy, 2015; Nogueira, 2017a,b, 2018). At
faster time-scales intermittency becomes non-negligible and the quasi-Gaussian
approximation to the statistics is no longer robust.
The proposed downscaling methodology corresponds to treating the sub-yearly
frequencies as random 'weather noise', which is characterized, to a good approximation,
by scale-invariant behavior with quasi-Gaussian statistics (Vallis, 2009; Lovejoy et al.,
2015). A similar downscaling methodology has been previously demonstrated to
reproduce the spatial sub-grid scale variability of topographic height (Bindlish & Barros,
1996), rainfall (Bindlish & Barros, 2000; Rebora et al., 2006; Nogueira & Barros, 2015)
and clouds (Nogueira & Barros, 2014).
Figure 5a showed that the PDFs computed from $\Delta P_{1m,DO}$, $\Delta P_{1m,Pr}$ and $\Delta P_{1m,SST}$ were in
remarkable agreement with GPCP PDFs for the 1979-2010 period, representing a
significant improvement compared to all atmospheric model-based products. This



improved PDF accuracy was quantified using the Perkins skill score, S-Score (Perkins et
al., 2007), defined as:
$$\text{S-Score} = 100 \times \sum_{i=1}^{M} min[f_{mod}(i), f_{obs}(i)] \qquad (12)$$
where $f_{mod}(i)$ and $f_{obs}(i)$ are respectively the frequency of the modeled and observed P
anomaly values in bin i, M is the number of bins used to compute the PDF (here M=15),
and min[x,y] is the minimum between the two values. The S-Score is a measure of
similarity between modeled and observed PDFs, such that if a model reproduces the
observed PDF perfectly then S-Score=100%.
The linear-based models showed S-Score values around 95%, which were significantly
higher than then ~80% found for the atmospheric model-based products (Fig. 6).
Furthermore, the stochastic model captured the change in the PDFs between two separate
periods (1979-1990 and 1999-2010, Fig. 5b), while preserving the remarkably high
(≥90%) S-Scores (Fig. 6, blue and red markers). Indeed, the S-Scores for linear-based
were consistently better than the S-Scores obtained from atmospheric model-based
products (~80%). Despite some differences between PDFs obtained from $\Delta P_{1m,DO}$,
$\Delta P_{1m,Pr}$ and $\Delta P_{1m,SST}$, their similar performance in reproducing observations was
somewhat unexpected, given the distinct performances in reproducing the observed time-
series at multi-year resolutions. While the error analysis here was based on a limited
sample (mainly due to short duration of the satellite-period), these results suggested that
the proposed stochastic downscaling mechanism is quite robust in reproducing the
monthly P statistics, with only moderate sensitivity to the coarse resolution forcing.

**5.   Conclusions**
Atmospheric variables display significant variability over a wide range of temporal
scales, both due changes in external forcings (including surface fluxes, changes to
greenhouse gases and aerosol concentrations, solar forcing, and volcanic eruptions), but
also due to intrinsic modes of atmospheric variability. Additionally, external and internal
atmospheric processes interact nonlinearly amongst themselves, resulting in an intricate
multi-scale structure, which is still ill understood and responsible for significant
uncertainties in climate models. Here a multi-scale analysis framework was employed,
aiming to disentangle the complex structure of global-averaged precipitation variability.
A critical transition emerges from DCCA at time-scales ~1-year, revealing a change in
the control mechanisms of the P and W, but also in the strength of the atmosphere-ocean



coupling. At multi-year time-scales W becomes tightly correlated to $T_{2m}$ and to SST
(~0.9), while at sub-yearly time-scales this correlation decreases abruptly towards
negligible values (~0.2). A sensitivity coefficient for W close to the typically estimated
0.07%/K was found for multi-year time-scales. In other words, the C-C relationship is the
dominant mechanism of W at multi-year time-scale, but not at sub-year time-scales.
Furthermore, at time-scales >1-2 years SST becomes tightly correlated to $T_{land}$, pointing
to a fundamental behavioral transition, where the atmosphere and the oceans start to act
as a single coupled system at multi-year time-scales, as previously suggested by Lovejoy
et al. (2018).
A similar transition was also found for $\rho_{P,T_{2m}}$ and $\rho_{P,SST}$, with negligible correlations and
sub-year time-scales, which tend increase at multi-year time-scales, although the latter
displayed significant spread amongst different datasets (ranging between ~0.4 to ~0.7).
More robust correlations were obtained for the P response to the energetic constraints
imposed by a simple atmospheric energy balance. DCCA showed that P variability is
tightly (negatively) coupled to the net atmospheric radiative balance at multi-year time-
scales (with $\rho_{P,R_{atm}} \lesssim -0.8$). The transition between multi-year and sub-yearly time-
scales also emerged for $\rho_{P,R_{atm}}$, with the correlation magnitude decreased rapidly at sub-
yearly time-scales, changing signal, and reached values ~0.4 at sub-monthly time-scales.
Additionally, DCCA revealed that the positive sub-monthly correlations are dominated
by the TOA OLR, while the multi-year correlations were dominated by surface longwave
fluxes, particularly by DLR. Furthermore, DCCA suggested that cloud effects play a
negligible on the multi-year correlations, but they are important for the sub-monthly
$\rho_{P,R_{atm}}$ values. Notice that the sensitivity coefficients of P to SST estimated here were in
the 2-4%/K range, close to the typical 1-3%/K values (for P against $T_S$) obtained from
energetic constraints on global rainfall.
The robustness and relevance of this emergent multi-scale correlation structure is
demonstrated by proposing simple models for reconstruction of P at multi-year time-
scales. Anomaly time-series for P at 2-year resolution were derived from SST
observations alone, either directly based on $\rho_{P,SST}$, or by combining $\rho_{R,DLR_{CS}}$, empirical
algorithms for clear-sky DLR estimation, and the C-C relationship. After correcting for
their systematic mean bias, the highly-idealized model for $\Delta P_{2y}$ based on clear-sky DLR
estimated from the Prata algorithm displayed similar accuracy in reproducing
observations as atmospheric model-based products, as measured by $RMSE_{bc}$, Pearson



correlation coefficient and normalized standard deviation. The simple model based on the
Dilley-O'Brien algorithm for clear-sky DLR estimation showed a somewhat poorer
performance, particularly in reproducing the observed variability amplitude. Finally, the
model based on P-SST correlation showed the weakest performance, which agrees with
the less robust correlations underlying this idealized model.
The proposed linear models cannot be extended to sub-yearly the time-scales because all
the correlations upon which they rely become negligible. This abrupt transition in the
multi-scale correlation structure implies that at sub-yearly time-scales the tight linear
coupling between atmospheric and ocean temperature, the Clausius-Clapeyron
relationship, and the atmospheric energy balance are no longer dominant linear
constraints for P. Nonetheless, the multi-scale analysis framework provides another path
for reconstruction of the P statistics at sub-yearly resolution. A stochastic downscaling
algorithm based on scale-invariant symmetries of P was applied to $\Delta P_{2y}$ reconstructed
time-series, resulting in monthly P PDFs. Remarkably, the reconstructed PDFs of P at
monthly resolution showed better accuracy in reproducing GPCP statistics than
atmospheric model-based products, as measured by S-Score computed over decadal and
30-year periods. Interestingly, the PDFs obtained by downscaling the three algorithms
proposed for multi-year P reconstruction showed similar performance, suggesting a weak
sensitivity of this algorithm to the accuracy of the 2-year resolution forcing time-series.
The present investigation highlights the complex multi-scale structure of the water cycle
variability and its governing mechanisms. Finally, it is hypothesized that the path for
stochastic regional precipitation simulation may be opened by leveraging on the widely
reported scale-invariant properties of precipitation in the spatial domain (e.g. Lovejoy a&
Schertzer, 2013; Nogueira & Barros, 2014, 2015), and exploring control mechanisms for
slow variability of regional precipitation, such as the El-Niño Southern Oscillation and
its teleconnections.

**Acknowledgements**
This study was funded by the Portuguese Science Foundation (F.C.T.), under grant
UID/GEO/50019/2013, as part of research project SOLAR
(PTDC/GEOMET/7078/2014).
ERA-20C and ERA-20CM were provided by ECMWF and are available through the
website http://apps.ecmwf.int/datasets.



20CR reanalysis, GISSTEMP and GPCP precipitation product were provided by the
NOAA/OAR/ESRL PD, Boulder, Colorado, USA, from their website
http://www.esrl.noaa.gov/psd.
The CERES-EBAF data were obtained from the NASA Langley Research Center
Atmospheric Science Data Center, from their website
https://eosweb.larc.nasa.gov/project/ceres/ebaf_surface_table

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



**Table 1** Linear regression coefficient $\alpha_{W,SST}$ estimated from $\Delta W/W_c$ against $\Delta SST$ at 2-
year resolution, assuming a relationship as given by Equation (1). The respective
coefficient of determination is also provided. The $\alpha_{W,SST}$ are computed for ERA-20C,
20CR, and for the ensemble of ERA-20CM simulations. Additionally, the coefficient is
estimated by pooling together ERA-20C, ERA-20CM (ensemble) and 20CR datasets.

| Dataset | $\alpha_{W,SST}$ $[K^{-1}]$ | $R^2$ |
|---|---|---|
| ERA-20C | 0.09 | 0.97 |
| 20CR | 0.10 | 0.92 |
| E20CM (Ensemble) | 0.07 | 0.92 |
| All Datasets | 0.08 | 0.91 |



**Table 2**. Linear regression coefficient $\alpha_{P,DLR}$ estimated from $\Delta P/P_c$ against $\Delta DLR$ at 2-
year resolution, assuming a relationship as given by Equation (11). The respective
coefficients of determination are also provided. The $\alpha_{P,DLR}$ values are computed for ERA-
20C, 20CR, and for the ensemble of ERA-20CM simulations. Additionally, the
coefficient is estimated by pooling together all datasets, including GPCP observations
against DLR from CERES-EBAF.

| Dataset | $\alpha_{P,DLR}$ $[(Wm^{-2})^{-1}]$ | $R^2$ |
|---|---|---|
| ERA-20C | 0.005 | 0.88 |
| 20CR | 0.003 | 0.51 |
| E20CM (Ensemble) | 0.004 | 0.81 |
| All datasets (includes observations) | 0.004 | 0.70 |



**Table 3**. Linear regression coefficient $\alpha_{P,SST}$ estimated from $\Delta P/P_c$ against $\Delta SST$ at 2-
year resolution. The respective coefficients of determination are also provided. The $\alpha_{P,SST}$
values are computed for ERA-20C, 20CR, for the ensemble of ERA-20CM simulations,
and for GPCP against ERA-20CM control SST forcing. Additionally, the coefficient is
estimated by pooling together all datasets.

| Dataset | $\alpha_{P,SST}$ $[K^{-1}]$ | $R^2$ |
|---|---|---|
| ERA-20C | 0.04 | 0.89 |
| 20CR | 0.02 | 0.35 |
| E20CM (Ensemble) | 0.02 | 0.73 |
| GPCP | 0.04 | 0.42 |
| All datasets (includes observations) | 0.02 | 0.53 |



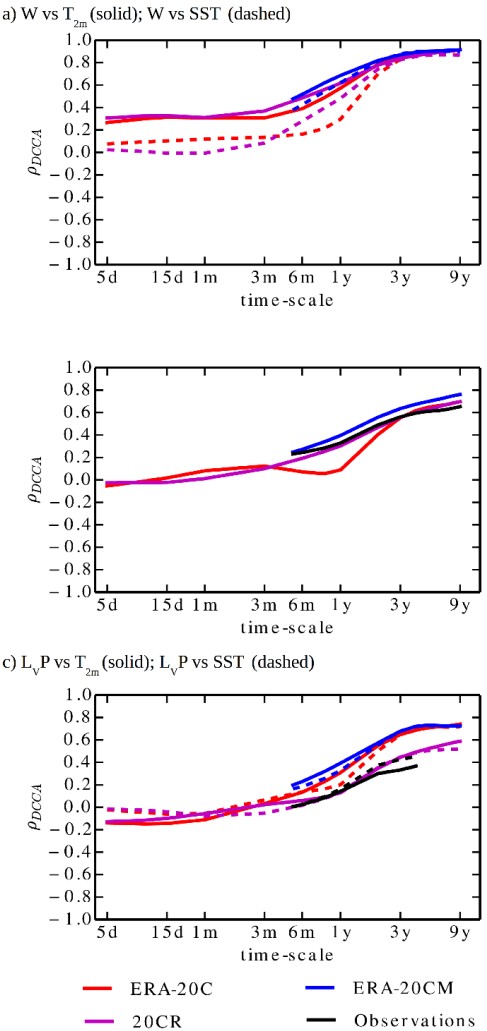


**Figure 1.** DCCA cross-correlation coefficients against temporal scale computed for global-mean time-series of a) $W$ vs $T_{2m}$ (solid) and $W$ vs $SST$ (dashed); b) $SST$ vs $T_{land}$; and c) $L_v P$ vs $T_{2m}$ (solid) and $L_v P$ vs $SST$ (dashed). Red lines represent results from ERA-20C, blue lines are from ERA-20CM, pink lines are from 20CR and black lines are estimated from observational products. Notice that $R_{atm}$ is not available from 20CR dataset, and that observational-based estimates of $\rho_{P,T_S}$ (and $\rho_{P,SST}$) are only computed up to 4-year time-scales due to the limited duration of GPCP dataset.

764

765

766





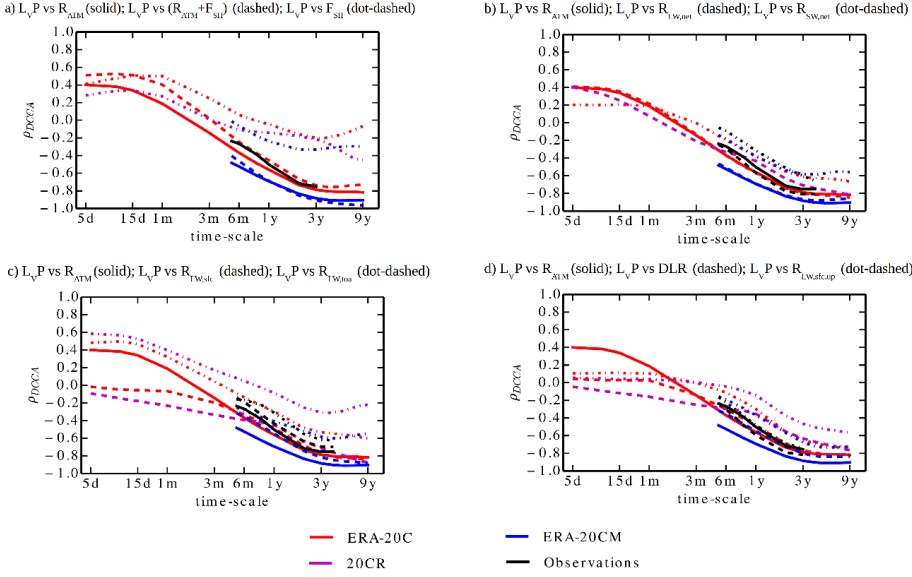

**Figure 2.** DCCA cross-correlation coefficients against temporal scale computed for a) $L_vP$ vs $R_{atm}$ (solid), $L_vP$ vs $(R_{atm} + F_{SH})$ (dashed) and $L_vP$ vs $F_{SH}$ (dot-dashed); b) $L_vP$ vs $R_{atm}$ (solid), $L_vP$ vs $R_{LW,net}$ (dashed), and $L_vP$ vs $R_{SW,net}$ (dot-dashed); c) $L_vP$ vs $R_{atm}$ (solid), $L_vP$ vs $R_{LW,SFC}$ (dashed), and $L_vP$ vs $R_{LW,TOA}$ (dot-dashed); and d) $L_vP$ vs $R_{atm}$ (solid), $L_vP$ vs $DLR$ (dashed), and $L_vP$ vs $R_{LW,SFC,UP}$ (dot-dashed). Red lines are computed from ERA-20C, blue lines are from ERA-20CM, pink lines are from 20CR and black lines are computed from GPCP and CERES-EBAF observational products. Notice that $R_{atm}$ and $R_{SW,net}$ are not available from 20CR, and that correlation coefficients estimated from observational products are only computed up to 4-year time-scales due to the limited duration of GPCP dataset.



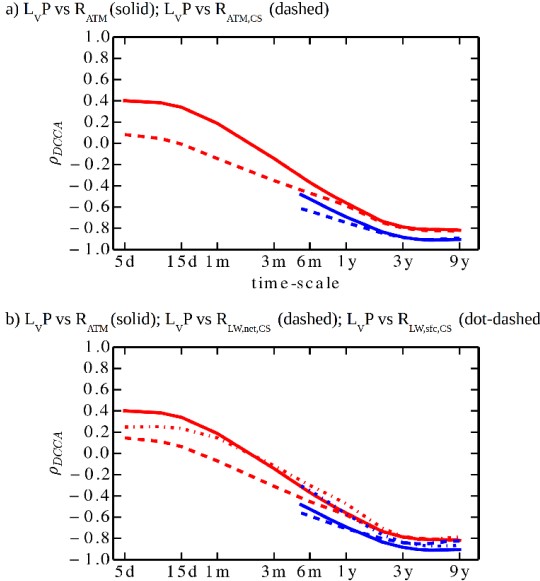

778

**Figure 3.** DCCA cross-correlation coefficients against temporal scale computed for a)
$L_vP$ vs $R_{atm}$ (solid) and $L_vP$ vs $R_{atm,CS}$ (dashed); b) $L_vP$ vs $R_{LW,SFC}$ (solid) and $L_vP$ vs
$R_{LW,SFC,CS}$ (dashed). Red lines are computed from ERA-20C and blue lines are from
ERA-20CM.

783





784

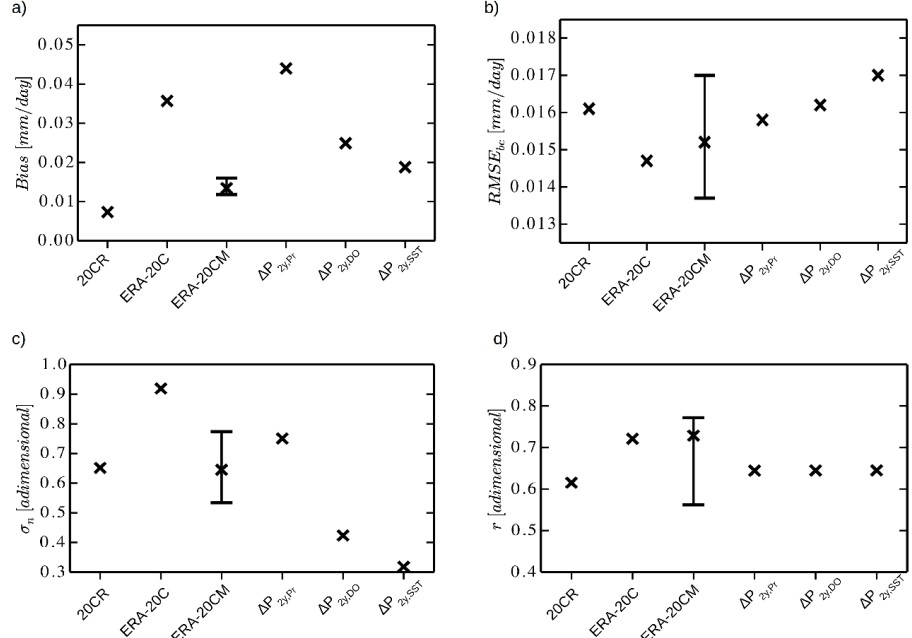

785

**Figure 4.** Error estimates from simulated anomaly time-series for P at 2-year resolution against GPCP, computed for the 1979-2010 period, including a) mean bias (Bias); b) root-mean-square error after bias correction ($RMSE_{bc}$); c) model standard deviation normalized by observed standard deviation ($\sigma_n$); and d) Pearson correlation coefficient (r). For ERA-20CM dataset the range for all ensemble members is shown, while 'x' marks their mean value. The p-value for all correlations shown in panel (d) are <0.05.

792





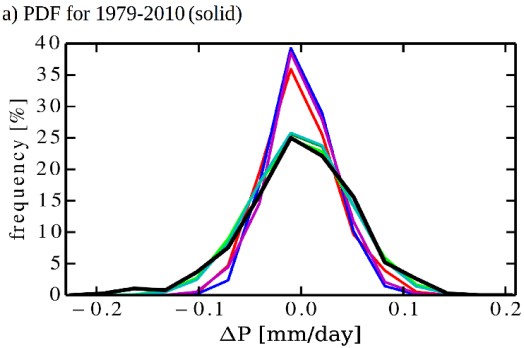

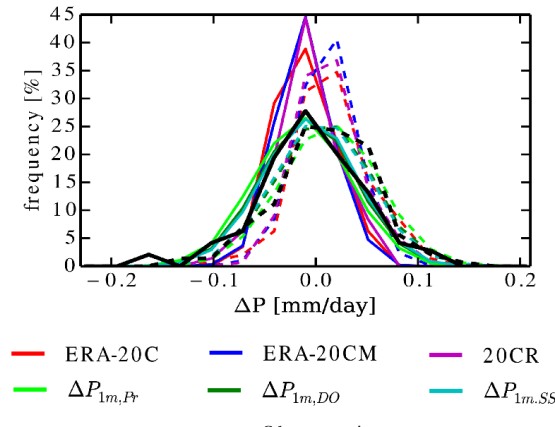

**Figure 5.** PDFs estimated from monthly anomaly time-series for P from ERA-20C (red), ERA-20CM (dark blue), 20CR (pink), GPCP (black), $\Delta P_{1m,DO}$ (dark green), $\Delta P_{1m,Pr}$ (light green), and $\Delta P_{1m,SST}$ (light blue). In panel a) the PDFs are estimated for the 1979-2010 period, and in panel b) the PDFs are estimated for the 1979-1990 period (solid) and the 1999-2010 period (dashed).

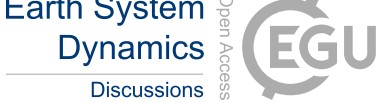



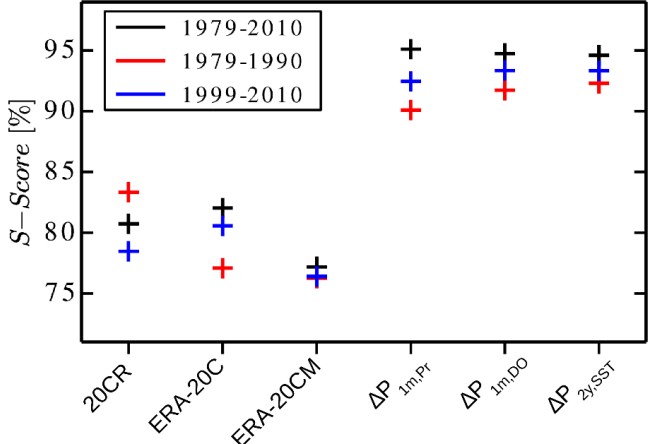

801

**Figure 6.** S-Score computed from the different P simulations against GPCP. The values
estimated for the full satellite period (1979-2010) are presented in black, for the 1979-
1990 period are presented in red, and for 1990-2010 period are presented in blue. For
ERA-20CM dataset, the S-Score is estimated from the 10-member ensemble PDF.