# Peer review of "An emergent transition time-scale in the atmosphere and its implications to global-averaged precipitation control mechanisms, time-series reconstruction and stochastic downscaling Miguel Nogueira\"

_Earth System Dynamics, 2018_

## Referee Comment (RC1) · Anonymous Referee #1 · 23 Oct 2018

I've found the reading of this paper very interesting and revealing. I am an engineering-hydrologist and the problem tackled in this paper is somehow far from what I am usually looking at, which is in general at a more local spatial and temporal scales. Nevertheless, it is interesting for us engineering-hydrologists to better understand the connections between Clausius-Clapeyron and precipitation, since C-C is sometimes used to explain changes even for local and extreme rainfall. I'm not an expert on climate dynamics and stochastic climate, therefore my review has to be taken as an outsider evaluation of the work. I hope the Author will find my remarks useful, although sometime naif, if he is interested to reach a wider readership among non-experts like myself.

Since I very much enjoyed reading the paper, as said, I have no major suggestions to provide, except the following two:

1) Title: based on my reading, the emergent transition timescale is not an original finding of this paper. The paper rather demonstrates the relevance of including the energy constraints of Equation (2), the atmospheric energy balance, to understand the elasticity of global precipitation to covariates. Shouldn't the title reflect this main focus of the paper?

2) It is unclear to me, at least it is not evident, why would one need the model proposed in Section 4 of the paper. To do what? To investigate how climate change may affect global precipitation? To build long-term global precipitation timeseries for the past? I would suggest the Author to be more clear on the usefulness of this part of the paper.

Detailed comments:

Line 22: I cannot figure out what is the "new perspective" provided by result (v). I would suggest the Author to be more explicit here.

Lines 92-93: is the improvement of climate simulations and future projections one of the final aims of the research conducted in this and related papers?

Lines 94-104: the Author has already looked in previous publications at the topic in the title of this paper. Therefore my suggestion to focus in the title on what new is in this paper. Is the main contribution the one stated at lines 105-106?

Lines 183-184: here a moving window procedure is applied. Presumably, what is evaluated in subsequent windows is highly correlated in time (because of the moving windowing). Isn't this a nuisance in the methodology, for instance when calculating the covariance in equation (6)? My experience is that doing statistics on moving windows is trickier than on non-overlapping ones. Maybe the Author could add a line to comment on eventual difficulties here.

Lines 184-188: maybe it is just me, but I did not understand this "local trend" removal. Why is it needed? I can see that this is what makes the cross-correlation analysis different at different time scales, but I would suggest the Author to explain its meaning also in plain words, for the non-technical readership.

Lines 201-203: what is the null hypothesis against which correlation significance is desired? Isn't it possible to obtain the right values for correlation significance, e.g. through simulations? Why is it difficult to do? Is it because of the overlapping moving windows used in the procedure?

Line 235: is the "large spread" in the results obtained using different datasets?

Lines 252-254: would one obtain larger correlations at short timescales if time lags would be used between variables? I would suggest the Author to discuss the time lag issue, if relevant.

Line 261: I guess "Fig. 1a" should be "Fig. 2a" here.

Lines 311-following: I would suggest here in the summary paragraph to restate the full names together with the acronyms. This is just my personal preference. As a non-expert reader, I am overwhelmed by the acronyms at this point.

Lines 334-344: this is a very important part of the paper, i.e., what are the implications of the results obtained here. I would suggest this part to be extended and maybe moved to the discussion section. I cannot really understand what "the more fundamental transition in the atmosphere" is. Also, the discussion between fast and slow P sensitivities deserves more space and discussion, if this paper is really shading more light on them.

Section 4. Stochastic model: I miss the motivation on why stochastic modelling is needed and useful here. I guess to demonstrate the robustness of the regression-based models, as stated at line 550. I would suggest the Author to explicitly comment on the usefulness of the section at its beginning.

[Figure]

Equation (9): here the Stefan-Boltzmann equation comes in, making the linear model to predict P based on DLR very different from the one directly using SST. Are the differences in Figure 4 (for example) explainable based on this difference of the two models?

Line 416 and Figure 4a: why is there bias in the simple regression models proposed in this paper? Shouldn't the calibration of the regression coefficients remove the bias? I guess it is more complex than that but I would suggest the Author to explain what could be the causes of bias in the GPCP models and in the proposed ones.

Line 431: here I would have expected differences because in one model SST comes in through the Stefan-Boltzmann equation, i.e. in a very non linear way. I am clearly missing something here. The Author may try to guide better the non-expert reader as me by explaining how the different models result in the same Pearson correlation coefficient of Figure 4d.

Section 4.2. Stochastic model at the monthly timescale: again, I miss the motivation on why stochastic modelling is needed and useful here.

Line 491: the result here, i.e. the fact that the proposed regression based models once downscaled outperform the GPCP models, is indeed remarkable. What could be the reason for that? I expect the Author to "speculate" on this in the discussion section.

Figures: it would be nice for the reader to see a plot of the timeseries of global variables (for P, SST, DLR etc), say at the annual (or larger) timescale for many years and at the seasonal timescale for a shorter period, to inspect visually their shape and potential relationship.

Figure 1: the heading of the second panel is missing.

Figure 4: isn't the RMSEbc simply the square root of the variance of estimation? If so, why calling it RMSEbc?

---

## Referee Comment (RC2) · S. Lovejoy (Referee) · 22 Nov 2018

General Comments This paper addresses the important question of the consequences of anthropogenic warming on global precipitation. Following Lovejoy et al 2017 (incorrectly cited as Lovejoy et al 2018), the author uses the cross-correlations of fluctuations to show that there is a relatively abrupt transition from weak to strong correlations with a transition at scales of around 1-2 years. He applies and develops this idea to quantities related to the precipitation energy budget. He thus clarifies the variables that become correlated with precipitation and the time scales at which this occurs.

[Figure]

Without such clarity about the appropriate time scales, any relationships between variables will be either questionable or spurious. Systematic approaches such as those presented here are therefore urgently needed; this paper has the potential of being a major contribution to the field.

To date, a technical issue that is important in the macroweather regime has prevented clarity. The macroweather regime covers roughly the lifetime of planetary structures ($\approx$10 days) and continues up until scales dominated by anthropogenic warming – i.e. up to the climate scales (currently at 20- 30 years for the temperature, a little longer for precipitation). In the climate regime, fluctuations begin to increase rather than decrease with scale. It is the ill-appreciated fact that standard correlation analyses suffer from low frequency biases due to the climate variability. The mathematical issue is that when fluctuations increase with scale – as they do in the climate regime - then correlation functions have low frequency divergences. It would be worthwhile for the author to mention this since this motivates his avoidance of the problem by the use of fluctuations.

In order to determine the fluctuations and their correlations, the author uses detrended cross-correlation analysis (DCCA). The DCCA is an adaptation of the detrended fluctuation analysis (DFA) technique and suffers from the same drawbacks and limitations. These are unfortunate consequences of its ad hoc nature: the fluctuations in the DFA and DCCA are defined in an unnecessarily obscure and complex-to-analyze manner (in terms of RMS residuals from polynomial regressions to the running sum of the original series). The resulting obscure definition leads to an inability to fully exploit the information contained in the fluctuations. For example, typical published DFA analyses do not even bother to put units on their fluctuation function because the function has no simple meaning! Only slopes to lines on log-log plots are considered interesting so that almost all the information contained in the fluctuations themselves is effectively discarded.

In the present paper, there is a similar waste of information: only the correlation coefficients at different time scales are given. The obscure meaning of his fluctuations prevents the author from directly making statements about the coefficients of the linear relations that are obtained at each time scale. This is a pity. If the author had used Haar fluctuations (simply the difference of the averages of the data over the first and second halves of an interval), the interpretation would have been nearly trivial. The author could have fixed the time scale and then, at that scale, display meaningful and insightful scatter plots of fluctuations of one quantity against another including linear (or other) regression relationships. By comparing plots at say $\Delta t$ = 1 month, 1 year, 10 years, one could then visually notice that the regression lines tighten up at larger $\Delta t$ and one could directly note the physically significant slopes at the longer (highly correlated) scales. One could then use standard statistical goodness of fit criteria and uncertainty estimates for the resulting regressions (correlation coefficients are not optimal for uncertainty analyses). Another advantage is that the author could also use multiple regression - simultaneously between fluctuations of several variables. At the moment, he is forced to make a series of awkward sequential tests of fluctuation pairs, trying to find the most significant relationships.

While the author's main conclusions are likely to be similar, the resulting paper would be more accessible and convincing. At the moment, most atmospheric science readers will simply see the DCCA as a "black box" and fail to appreciate its significance.

My appreciation "major revision" is given only in order to encourage the author to make further improvements so that his paper will have greater impact.

Detailed comments: 1) The number of acronyms was enormous and I was constantly searching through the text to remind myself of the more obscure ones. Perhaps the author could provide a convenient table for this purpose? 2) The fGn simulation is obtained by filtering Gaussian white noise. In principle this is fine, but there are potential high and low frequency numerical issues and it would probably worth using a packaged routine (available now on a number of platforms).

-Shaun Lovejoy

---

## Author Comment (AC1) · 27 Dec 2018

Author Comments on Review by Anonymous Reviewer #1

I've found the reading of this paper very interesting and revealing. I am an engineering hydrologist and the problem tackled in this paper is somehow far from what I am usually looking at, which is in general at a more local spatial and temporal scales. Nevertheless, it is interesting for us engineering-hydrologists to better understand the connections between Clausius-Clapeyron and precipitation, since C-C is sometimes used

to explain changes even for local and extreme rainfall. I'm not an expert on climate dynamics and stochastic climate, therefore my review has to be taken as an outsider evaluation of the work. I hope the Author will find my remarks useful, although some time naif, if he is interested to reach a wider readership among non-experts like myself.

R: I want to thank Reviewer #1 for his detailed, very useful and timely review of the manuscript. Responses to all Reviewers' comments are provided below. All changes to the original manuscript are highlighted in yellow in the main document.

Major comments:

Title: based on my reading, the emergent transition timescale is not an original finding of this paper. The paper rather demonstrates the relevance of including the energy constraints of Equation (2), the atmospheric energy balance, to understand the elasticity of global precipitation to covariates. Shouldn't the title reflect this main focus of the paper?

R: This is a good point. I've changed the title to "The multi-scale structure of the atmospheric energetic constraints on global-averaged precipitation" which should better reflect the main focus of the present manuscript.

It is unclear to me, at least it is not evident, why would one need the model proposed in Section 4 of the paper. To do what? To investigate how climate change may affect global precipitation? To build long-term global precipitation timeseries for the past? I would suggest the Author to be more clear on the usefulness of this part of the paper.

R: After re-reading Section 4 and the conclusions on this Section with the this comment in mind I agree that this point is not clear in the manuscript. The model has two separate parts: the first is a direct application of a linear response of multi-year precipitation fluctuations to fluctuations in the atmospheric radiative fluxes (or temperature). This linear relation is suggested by the respective strong correlations found at multi-year time-scales. The main goal of employing this linear model is to demonstrate

the validity of the correlations reported and how they are directly translated in a direct climate response (sensitivity) of precipitation to radiative fluxes (or temperature): for example, a fluctuation in DLR at multi-year time-scale has a direct response of precipitation which is, to a very good approximation, linear. The second part is based on the multi-scale stochastic scale-invariant properties of fields. The scale-invariant properties of precipitation and other atmospheric also show transition at a similar range of time-scales ($\sim$10-days to 1-month in the atmosphere and $\sim$1-year in the oceans, references in the manuscript), separating two different scale-invariant scaling regimes. Thus, the stochastic scale-invariance should be intrinsically connected to the correlation structure emerging from the results. Furthermore, stochastic scale-invariance has very high potential for stochastic downscaling applications, since it establishes simple relations between the statistics at different time-scales. This potential is demonstrated by the present results in this manuscript. To make the messages clearer, I've split Section 4 in two Sections (4 and 5), the first presenting the linear fluctuation model results and the second presenting the stochastic scale-invariant downscaling. I've also added an introductory paragraph to each of these Sections and edited the abstract and conclusions to make the message clearer.

Minor Comments:

Line 22: I cannot figure out what is the "new perspective" provided by result (v). I would suggest the Author to be more explicit here.

R: There is large spread in the estimates of precipitation sensitivity to temperature fluctuations, for example amongst CMIP5 models. The spread in the correlations here between different datasets also suggest spread in precipitation sensitivity to temperature fluctuations. However, I recognize this is not this simple and, additionally, not really explored in the present manuscript so I decided to remove this sentence.

Lines 92-93: is the improvement of climate simulations and future projections one of the final aims of the research conducted in this and related papers?

R: As stated in the beginning of the introduction "even the long-term response of global-averaged precipitation is still poorly understood, constrained and simulated (Collins et al., 2013; Allan et al., 2014; Hegerl et al., 2015), largely due to the limited knowledge on the complex interactions between the key components of the atmospheric branch of the water cycle and its forcing mechanisms.". The line of work presented here allows to disentangle some this complexity, to evaluate how models reproduce the observed variability and the key mechanisms controlling the variability at different time-scales. This is the idea of this sentence.

Lines 94-104: the Author has already looked in previous publications at the topic in the title of this paper. Therefore my suggestion to focus in the title on what new is in this paper. Is the main contribution the one stated at lines 105-106?

R: Good point, the title was changed accordingly.

Lines 183-184: here a moving window procedure is applied. Presumably, what is evaluated in subsequent windows is highly correlated in time (because of the moving windowing). Isn't this a nuisance in the methodology, for instance when calculating the covariance in equation (6)? My experience is that doing statistics on moving windows is trickier than on non-overlapping ones. Maybe the Author could add a line to comment on eventual difficulties here.

R: This is an interesting and difficult question, which is not directly answered here. First, notice that at the suggestion of the Reviewer, the multi-scale analysis was changes from DCCA to Haar fluctuations. Nonetheless, the correlations of Haar fluctuations are also obtained considering overlapping windows. I've added a note of caution at the end of Section 2.2 on this point. The key argument is that the present investigation uses a previously developed and demonstrated methodology (Haar fluctuation correlations) and assumes it accurately represents the correlation structure. Additionally, it is shown that the Haar fluctuation correlations are identical to the cross-correlations derived using DCCA (another well-established methodology DCCA, see Section 3). Thirdly, the
derived correlations have some physical meaning (energetic constraints of precipitation, Clausius-Clapeyron,…). Furthermore, Podobnik et al. (2011) have compared overlapping and non-overlapping windows in DCCA and established the significance of both. Finally, a more empirical argument is that overlapping boxes are a widely used method that allows us to obtain betters statistics because the data points are finite.

Lines 184-188: maybe it is just me, but I did not understand this "local trend" removal. Why is it needed? I can see that this is what makes the cross-correlation analysis different at different time scales, but I would suggest the Author to explain its meaning also in plain words, for the non-technical readership.

R: This comment by the Reviewer highlights the importance of the major revision suggested by the other Reviewer, changing DCCA to Haar fluctuations. It is true that DCCA is not very transparent. At Reviewer #2 suggestion I've changed the multi-scale correlation estimation methodology from DCCA to Haar fluctuations, which I believe are easier to understand. Nonetheless, the key idea in both methods is to disentangle the fluctuations at a given time-scale (and repeat the procedure at a wide range of scales). Removing the local trend is, in plain-words, removing lower-frequency variability in a nonstationary time-series.

Lines 201-203: what is the null hypothesis against which correlation significance is desired? Isn't it possible to obtain the right values for correlation significance, e.g. through simulations? Why is it difficult to do? Is it because of the overlapping moving windows used in the procedure?

R: Podobnik et al. (2011) developed and employed two statistical tests for the significance of DCCA cross-correlations as function of time lag n, based on the assumption that a series is either uncorrelated or power-law correlated. They show that to test the significance of DCCA cross-correlation and reject the null hypothesis it is necessary to compare it with a critical point, which are obtained using the detrending approach depends on the level of confidence required (e.g. 95%), time-series length and length

of the overlapping window considered. For the goal of the present paper, it seems to me that it is sufficient to see that there is a clear transition in the correlation magnitudes between time-scales, from strong to weak. The weak correlations are within the values previously shown to be negligible for this type of methods. Adding further complication and computing the critical points for my case, would make the paper more obscure without a clear gain in my opinion. I've tried to make this point clearer in the methodology Section and refer to this previous work.

Line 235: is the "large spread" in the results obtained using different datasets?

R: You are correct, the sentence was reworded.

Lines 252-254: would one obtain larger correlations at short timescales if time lags would be used between variables? I would suggest the Author to discuss the time lag issue, if relevant. R: No. The correlations at different time-lags was tested in Nogueira (2018) and no relevant change of the correlation structure was found. This reference was included in the text.

Line 261: I guess "Fig. 1a" should be "Fig. 2a" here. R: Yes. This was corrected, thank you

Lines 311-following: I would suggest here in the summary paragraph to restate the full names together with the acronyms. This is just my personal preference. As a non-expert reader, I am overwhelmed by the acronyms at this point.

R: I have removed most of the acronyms throughout the manuscript, replacing with full names, to make the text clearer.

Lines 334-344: this is a very important part of the paper, i.e., what are the implications of the results obtained here. I would suggest this part to be extended and maybe moved to the discussion section. I cannot really understand what "the more fundamental transition in the atmosphere" is. Also, the discussion between fast and slow P sensitivities deserves more space and discussion, if this paper is really shading more

light on them.

R: I have moved this discussion to the final section and elaborated on how the correlation structure relates to the fast and slow components. The "more fundamental transition" refers to ocean-atmosphere coupling as seen by SST vs Tland correlations. This is also discussed in the final section.

Section 4. Stochastic model: I miss the motivation on why stochastic modelling is needed and useful here. I guess to demonstrate the robustness of the regression based models, as stated at line 550. I would suggest the Author to explicitly comment on the usefulness of the section at its beginning.

R: I've separated the stochastic model in Section 5 and added a paragraph in the beginning to explain its motivation.

---

## Author Comment (AC2) · 27 Dec 2018

Author Comments on Review by Shaun Lovejoy

This paper addresses the important question of the consequences of anthropogenic warming on global precipitation. Following Lovejoy et al 2017 (incorrectly cited as Lovejoy et al 2018), the author uses the cross-correlations of fluctuations to show that there is a relatively abrupt transition from weak to strong correlations with a transition at scales of around 1-2 years. He applies and develops this idea to quantities related

to the precipitation energy budget. He thus clarifies the variables that become correlated with precipitation and the time scales at which this occurs. Without such clarity about the appropriate time scales, any relationships between variables will be either questionable or spurious. Systematic approaches such as those presented here are therefore urgently needed; this paper has the potential of being a major contribution to the field.

R: I want to thank Shaun Lovejoy for his insightful review, which I believe has helped to improve the quality of the manuscript. Responses to all Reviewers' comments are provided below. The Reviewer's comments are in black and the author's responses are in blue. The citation Lovejoy et al. (2018) was corrected to Lovejoy et al. (2017), thank you. All changes to the original manuscript are highlighted in yellow in the main document.

To date, a technical issue that is important in the macroweather regime has prevented clarity. The macroweather regime covers roughly the lifetime of planetary structures (_10 days) and continues up until scales dominated by anthropogenic warming – i.e. up to the climate scales (currently at 20- 30 years for the temperature, a little longer for precipitation). In the climate regime, fluctuations begin to increase rather than decrease with scale. It is the ill-appreciated fact that standard correlation analyses suffer from low frequency biases due to the climate variability. The mathematical issue is that when fluctuations increase with scale – as they do in the climate regime – then correlation functions have low frequency divergences. It would be worthwhile for the author to mention this since this motivates his avoidance of the problem by the use of fluctuations.

R: This is a good point, I've included a mention to this issue in Section 2.2.

In order to determine the fluctuations and their correlations, the author uses detrended cross-correlation analysis (DCCA). The DCCA is an adaptation of the detrended fluctuation analysis (DFA) technique and suffers from the same drawbacks and limitations.

These are unfortunate consequences of its ad hoc nature: the fluctuations in the DFA and DCCA are defined in an unnecessarily obscure and complex-to-analyze manner (in terms of RMS residuals from polynomial regressions to the running sum of the original series). The resulting obscure definition leads to an inability to fully exploit the information contained in the fluctuations. For example, typical published DFA analyses do not even bother to put units on their fluctuation function because the function has no simple meaning! Only slopes to lines on log-log plots are considered interesting so that almost all the information contained in the fluctuations themselves is effectively discarded. In the present paper, there is a similar waste of information: only the correlation coefficients at different time scales are given. The obscure meaning of his fluctuations prevents the author from directly making statements about the coefficients of the linear relations that are obtained at each time scale. This is a pity. If the author had used Haar fluctuations (simply the difference of the averages of the data over the first and second halves of an interval), the interpretation would have been nearly trivial. The author could have fixed the time scale and then, at that scale, display meaningful and insightful scatter plots of fluctuations of one quantity against another including linear (or other) regression relationships.

R: Thank you, this is a good argument, which was corroborated by the doubts on the DCCA fluctuations and detrending technique raised by the other Reviewer. Hence, I've decided to use Haar fluctuations instead of DCCA. I've added the description of Haar fluctuations in Section 2.2, changed the results presentation in Section 3 accordingly (including Figures 1, 2 and 3), and made also made the respective changes to abstract and conclusions. As expected, the resulting correlation at different time-lags are remarkably identical to the results previously obtained by DCCA (now presented in supplementary material), providing robustness to the correlation structure presented. Since Haar fluctuations are easier to interpret, I decide to keep the latter.

By comparing plots at say Dt = 1 month, 1 year, 10 years, one could then visually notice that the regression lines tighten up at larger Dt and one could directly note the physically significant slopes at the longer (highly correlated) scales. One could then use standard statistical goodness of fit criteria and uncertainty estimates for the resulting regressions (correlation coefficients are not optimal for uncertainty analyses). Another advantage is that the author could also use multiple regression - simultaneously between fluctuations of several variables. At the moment, he is forced to make a series of awkward sequential tests of fluctuation pairs, trying to find the most significant relationships. While the author's main conclusions are likely to be similar, the resulting paper would be more accessible and convincing. At the moment, most atmospheric science readers will simply see the DCCA as a "black box" and fail to appreciate its significance.

R: After the major change to methodology, I decided to keep some of the manuscript structure. I believe that Figures 1, 2 and 3 (now obtained from correlations of Haar fluctuations) are good for highlighting and illustrating a relevant transition in the governing mechanisms of precipitation (and precipitable water vapor) mechanisms between sub-yearly and multi-year time-scales. The variables in these correlation plots are guided by Clausius-Clapeyron and radiative constraints of precipitation, and hence their choice is not random. The suggested scatter plots would in fact have relevant information, but this information is partly in the correlations and partly in the ability of the tested simple linear model in reproducing observations: i.e. the fluctuations in precipitation can in fact be derived from DLR (and somewhat worse from SST) fluctuations using a sensitivity coefficient. Adding the scatter plots would make the manuscript too extensive without too much added value, in my opinion.

The number of acronyms was enormous and I was constantly searching through the text to remind myself of the more obscure ones. Perhaps the author could provide a convenient table for this purpose?

R: I've reduced drastically the number of acronyms throughout the manuscript, replacing by the full names in most cases.

The fGn simulation is obtained by filtering Gaussian white noise. In principle this is fine, but there are potential high and low frequency numerical issues and it would probably worth using a packaged routine (available now on a number of platforms)

R: The fGn generation algorithm employed does not seem to have low or high frequency numerical issues for the considered. The Figure attached is the ensemble spectrum of the 100 realizations of beta=0.3 interpolating time-series (black) with the respective reference beta=0.3 line (red)
* * *
[Figure]

**Fig. 1.**